https://doi.org/10.1038/s41467-020-15114-1　　**OPEN**

# Predicting gene expression using morphological cell responses to nanotopography

Marie F.A. Cutiongco [1], Bjørn Sand Jensen[2], Paul M. Reynolds [1] & Nikolaj Gadegaard [1]✉

Cells respond in complex ways to their environment, making it challenging to predict a direct relationship between the two. A key problem is the lack of informative representations of parameters that translate directly into biological function. Here we present a platform to relate the effects of cell morphology to gene expression induced by nanotopography. This platform utilizes the 'morphome', a multivariate dataset of cell morphology parameters. We create a Bayesian linear regression model that uses the morphome to robustly predict changes in bone, cartilage, muscle and fibrous gene expression induced by nanotopography. Furthermore, through this model we effectively predict nanotopography-induced gene expression from a complex co-culture microenvironment. The information from the morphome uncovers previously unknown effects of nanotopography on altering cell–cell interaction and osteogenic gene expression at the single cell level. The predictive relationship between morphology and gene expression arising from cell-material interaction shows promise for exploration of new topographies.

---

[1] Divison of Biomedical Engineering, School of Engineering, University of Glasgow, Glasgow, UK. [2] School of Computing Science, University of Glasgow, Glasgow, UK. ✉email: Nikolaj.Gadegaard@glasgow.ac.uk

Biomedical implants continue to be developed to improve patient outcomes. One way to enhance implant efficacy and tissue regeneration is to vary substrate texture with nanotopographies. Topographies at the cell-material interface are widely shown to direct cell behavior: nanopillars change cell morphology[1]; nanogratings drastically alter lipid metabolism[2], and pluripotent[3,4] and multipotent cell differentiation[5]; and subtle changes to nanopit geometric arrangement switches human mesenchymal stem cells from multipotent to osteogenic fate[6–9]. Morphological responses to nanotopography are manifested through varying focal adhesion size, orientation and composition[10–14], and changes in actin contractility and nuclear deformation[15].

A quantitative relationship exists between a material's physicochemical structure and its biological activity. Rational drug design has long relied on molecule solubility, ionization and lipophilicity to predict activity[16]. Protein engineering has similarly modeled protein–peptide interactions from protein structure[17]. Cell metabolic activity correlates with synthetic polymer composition, glass transition temperature, and water contact angle[18]. Meanwhile, bacterial attachment can be predicted from descriptors of secondary ionic hydrocarbon chains[19]. In contrast to active biomolecules, the mechanotransductive effects of topography on cell response do not intuitively relate to topography length scale, isotropy, geometry, and polarity. This limits the discovery of functional topography to the screening of libraries for hits using a single, representative cell type[20–22]. Among its limitations (which include cost, inefficiency and the sampling of a small topography space), this screening approach disregards the cell specificity of response to nanotopogrpahy. Thus, it is vitally important to develop a systematic method to capture cell phenotypes (both at morphological and functional levels) induced by topography.

Here, we demonstrate an image profiling-based platform that encompasses morphological and functional responses induced by nanotopography. Single-cell measurements of focal adhesions, actin cytoskeleton and chromatin, referred to as the "morphome", clearly reflected cell type and nanotopography. Using the morphome as predictors and without prior knowledge about nanotopography or cell type, a Bayesian linear regression model robustly predicts quantitative gene expression levels induced by nanotopography. Excluding topography metrics (e.g., diameter, pitch of topography) in the model and instead relying on cell type-dependent mechanobiological responses, highlights the broad applicability of this platform to many biomaterial and cell systems. We have used the platform to understand single-cell gene expression in a co-culture environment of osteoblasts and fibroblasts. Not only could we confirm osteogenic gene expression, but the platform also provided us with insight into the interplay between individual cells on nanotopography. Thus, here we present a quantified and predictive relationship between morphology, gene expression, and topography at the single-cell level.

## Results

**Morphological cell responses: the morphome**. We used nanopit topographies consisting of 120 nm diameter, 100 nm depth and with a 300 nm center-to-center distance in a square array (SQ)[6,7], hexagonal array (HEX)[23,24], and arranged with center-to-center distance offset from 300 nm by 50 nm in both x and y directions (NSQ array)[6,7]. An unpatterned ("FLAT") surface was used as a control (Fig. 1a).

We employed cells of the musculoskeletal system due to the diverse responses of muscle, bone, cartilage, and fibrous cell types to nanotopographies[23–26]. Mouse myoblasts, osteoblasts,

chondrocytes, fibroblasts, and pre-osteoblast[27] and pre-myoblast[28] progenitors, were grown on nanotopographies. Responses from combinations of each cell type on all nanotopographies were measured, yielding 24 unique combinations of cell type and nanotopography. Effects of nanotopography on conventional morphological characteristics such as cell and nuclear area, actin intensity, focal adhesion area, and intensity (see Supplementary Figs. 1 and 2, and Supplementary Note 1), and nuclear translocation of the mechanosensors YAP and TAZ (see Supplementary Figs. 3 and 4) were evident. Quantitative polymerase chain reaction (QPCR) was then used to assess changes in lineage marker expression induced by nanotopography by day 7 (Fig. 1b). At day 2, we performed image-based profiling (Fig. 1c). From images of the chromatin and actin, the nucleus and the cell body, respectively, was robustly segmented. Within these cellular features, we measured morphology (shape and geometry of different compartments), texture (spatial patterns of fluorescence and therefore organization), intensity (total fluorescence value), and radial distribution of intensity (measuring radial arrangement of fluorescence) of chromatin, actin, focal adhesion kinase (FAK) and phosphorylated FAK (pFAK) (Fig. 1d). The morphome consisted of 624 single-cell measurements ("features"), with 75 chromatin and nuclear features, 211 actin and whole-cell features, 168 FAK features, and 170 pFAK features. Machine learning was then applied on the morphome: hierarchical clustering was used to uncover distinct patterns of morphological features that distinguish cell type-specific responses to nanotopographies (Fig. 1e); (ii) Bayesian linear regression was then used to predict myogenic, osteogenic, chondrogenic and fibrogenic gene expression induced by nanotopography using the morphome as predictors (Fig. 1f).

**The morphome captures changes induced by nanotopographies**. Patterns of nanotopography-induced morphological changes were visible from the morphome (Fig. 2). Immediately apparent were large blocks of actin, FAK, and pFAK measurements with similar values within a cell type on a specific nanotopography. These features correspond to increasingly complex measures of texture, granularity and radial intensity distribution for chromatin, actin, FAK and pFAK (Fig. 2b–f, see Supplementary Table 1). Frequency of pixel gray levels measure texture and homogeneity of pixels, with high values indicating coarseness. Granularity measures an object's coarseness, with higher values indicating heterogeneity of pixel intensities and coarser texture. The Zernike coefficient measures the spatial arrangement of intensity as it resembles the increasingly complex Zernike polynomials (Fig. 2f). The Zernike coefficient was used to measure both cell shape and radial distribution of fluorescence intensity of chromatin, actin, pFAK, and FAK. Interestingly, higher order Zernike polynomials resemble the punctate shape and spatial distribution of focal adhesions[29]. This provides an integrative analysis of focal adhesions at the single-cell level compared to traditional measures that define individual adhesion characteristics.

**Nanotopography changes gene expression**. Gene expression was used to quantitatively determine the effect of nanotopographies on cell function (see Supplementary Fig. 5 and Supplementary Data 2). For comparison, we differentiated the same cells cultured on conventional tissue culture plastic using established biochemical inducers of musculoskeletal differentiation (see Supplementary Methods). We discuss here the statistically significant changes induced by nanotopography on lineage-specific gene expression relevant to the cell type (see Supplementary Table 2). Pre-myoblasts showed significantly higher expression of the early

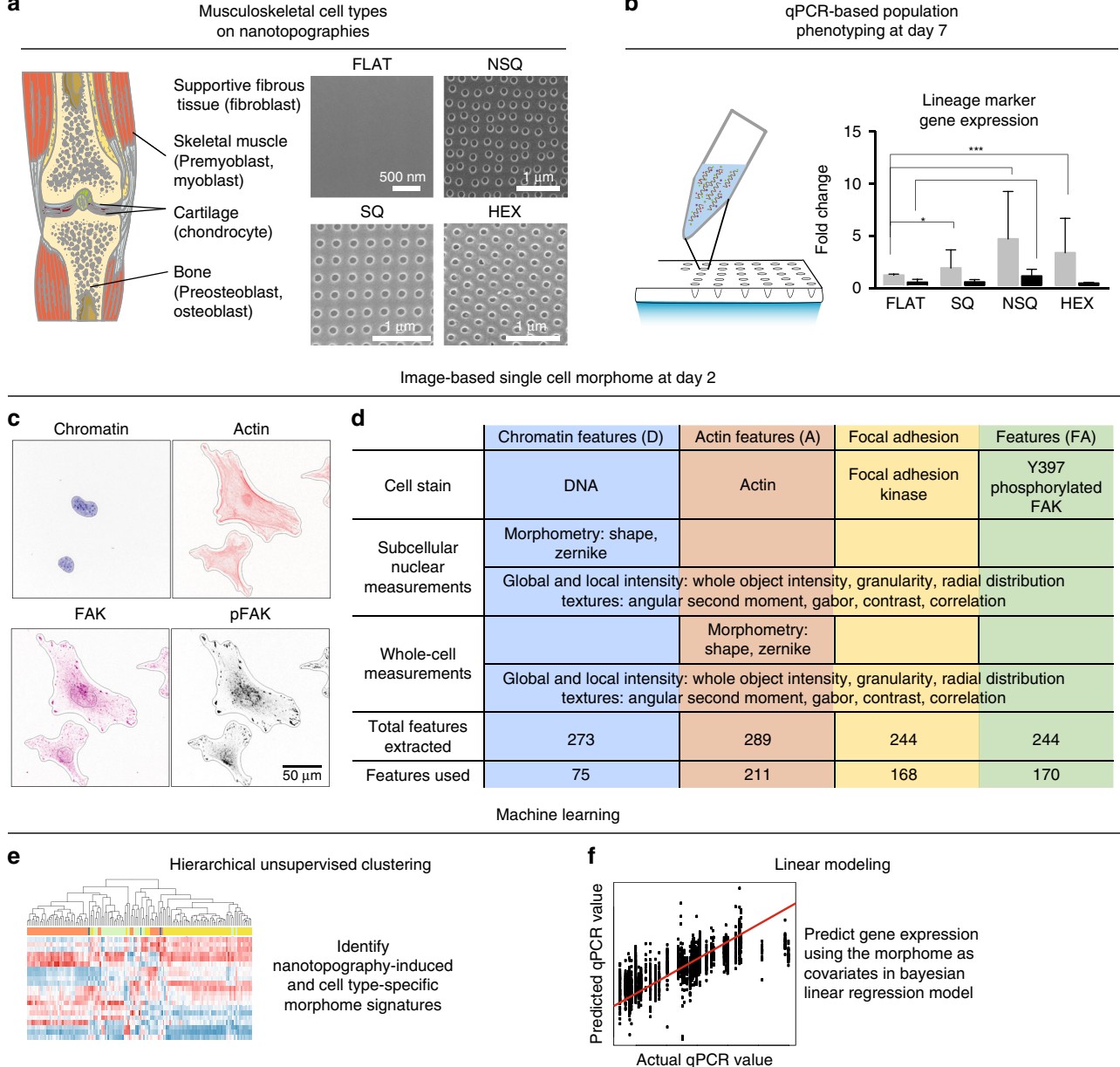

**Fig. 1 The morphome relates gene expression and morphology changed by nanotopography. a** Osteoblastic, myoblastic, chondroblastic, and fibroblastic cell lines were cultured on nanotopographies to obtain 24 combinations of cell type and topography. Precursor (pre-myoblast and pre-osteoblast) and lineage committed (myoblast and osteoblast) cells were from the same cell line, but with lineage committed cells cultured in the presence of inducers of myogenic or osteogenic differentiation. Image of musculoskeletal system obtained from Servier Medical Art under CC-BY 3.0. Servier Medical Art by Servier is licensed under a Creative Commons Attribution 3.0 Unported License (http://creativecommons.org/licenses/by/3.0/). **b** At day 7, lineage-specific gene expression induced by nanotopography was measured using population-based quantitative polymerase chain reaction (QPCR). **c, d** Image-based cell profiling. At day 2, various measures of chromatin (blue), actin (red), focal adhesion kinase (FAK, yellow), and phosphorylated FAK (pFAK, green) were obtained from images of single cells. Collectively, multivariate dataset containing single-cell measures of focal adhesions, the cytoskeleton, and chromatin is referred to as the "morphome". **c** Representative images of cells stained against different cellular aspects. Black lines show the cell and nucleus outlines, from which morphology measurements were extracted. Cell and nucleus outlines were obtained from the actin and chromatin images, respectively. **d** Morphome features extracted from 4 stains across single cells. **e**, **f** Machine learning for data-driven exploration and model building using the morphome. **e** Hierarchical clustering uncovered distinct patterns that delineate cell type and nanotopography within the morphome. **f** Bayesian linear regression created a predictive model that related the morphome to gene expression.

lineage marker *MYOD1*, and of the late markers *MYOG* and *MYH7* when cultured on SQ surfaces relative to FLAT surfaces (Fig. 3a, b). This myogenic gene expression profile was similar to pre-myoblasts stimulated with biochemical inducers of myogenic differentiation for 4 days (see Supplementary Fig. 6a, e). Both pre-osteoblasts and osteoblasts showed increased expression of early

(*RUNX2*, *SP7*) and late (*BGLAP*, *SPP1*) osteogenic markers when cultured on NSQ relative to FLAT (Fig. 3d–g), in line with previous studies[7–9]. The gene expression profile of both pre-osteoblasts and osteoblasts on NSQ resembled cells osteogenically differentiated after 4 days (see Supplementary Fig. 6b, f). Chondrocytes cultured on HEX showed increased expression of

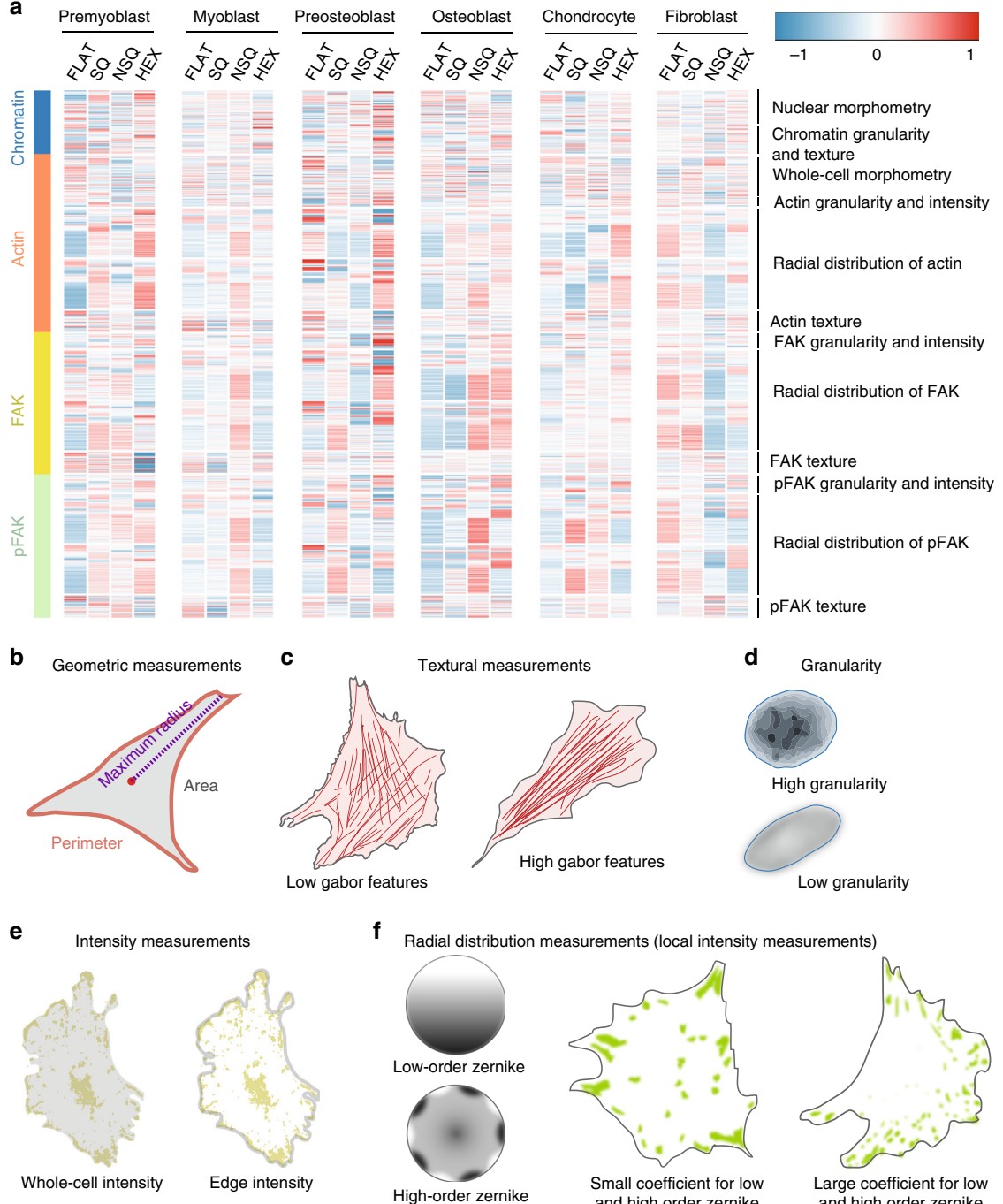

**Fig. 2 Cell response to nanotopography at single-cell level.** Osteoblastic, myoblastic, chondroblastic, and fibroblastic cell lines cultured on different nanotopographies were used to obtain the morphome. **a** Heat map of the morphome across cell types and nanotopographies. The morphome consisted of 624 features that quantitatively measure the cell and nucleus geometries, as well as chromatin, actin, FAK, pFAK characteristics within single cells (see Supplementary Data 1). Each feature has an average of 0 and a standard deviation of 1 after normalization across all cell types. The color and intensity of each tile represents the average value of the feature for a particular cell type and nanotopography combination. **b–f** Schematic diagrams of representative morphome features.

*COL2A1* (early marker) and *ACAN* (late marker) compared to those cultured on FLAT (Fig. 3h–k). Chondrogenic gene expression profile induced by SQ and HEX showed the highest similarity with cells chondrogenically differentiated for 4 days (see Supplementary Fig. 6c, g). Interestingly, this means that each nanotopography favors the gene expression of separate cell phenotypes. Meanwhile, fibroblasts showed increased expression of pathogenic fibrosis markers, *TGFB1I1*, *COL3A1*, and *ELN*[30,31] on

all nanotopographies compared with FLAT (Fig. 3l–n). However, we did not observe any similarities in fibrotic gene expression profile induced by nanotopographies and fibrotic induction (see Supplementary Fig. 6d, h).

**Distinct nanotopographies are reflected in the morphome.** A subset of the morphome, consisting of 185 features, varied

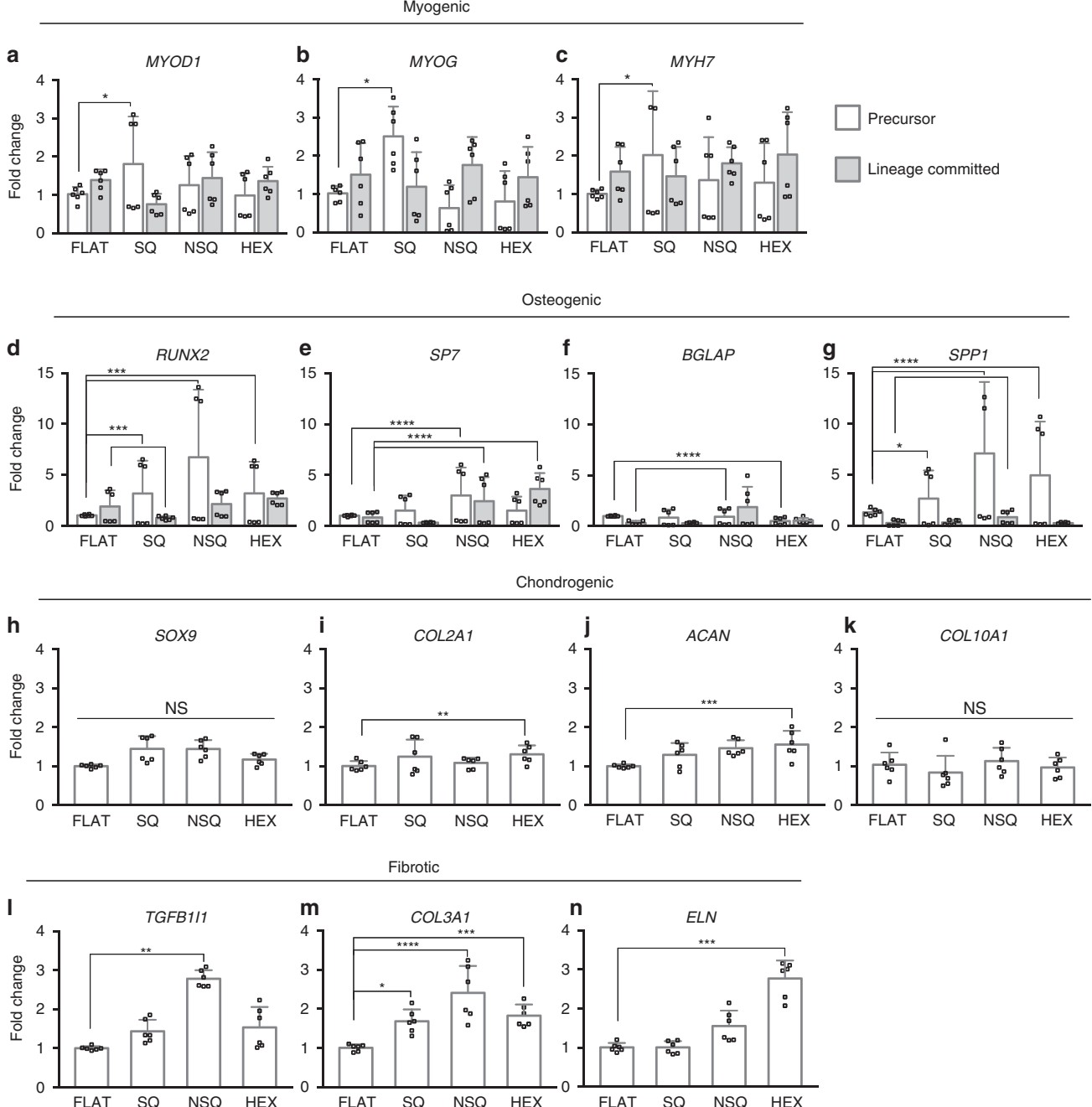

**Fig. 3 Gene expression is changed by nanotopography.** Changes in musculoskeletal gene expression from musculoskeletal cell types in response to nanotopographies. Measurement of **a–c** myogenic markers from pre-myoblasts and myoblasts, **d–g** osteogenic markers from pre-osteoblasts and osteoblasts, **h–k** chondrogenic markers from chondrocytes, **l–n** fibrotic markers from fibroblasts on nanotopographies. See Supplementary Data 2 for the complete gene expression data. Precursor (pre-myoblast and pre-osteoblast) and lineage committed (myoblast and osteoblast) cells were from the same cell line, but with lineage committed cells cultured in the presence of inducers of osteogenic or myogenic differentiation. Gene expression is listed in order of increasing maturity for the given cell lineage. QPCR measurements shown here were normalized to the reference gene and cell type on FLAT. All QPCR measurements are given as mean ± standard deviation from two independent experiments ($n = 6$). Open faced squares denote individual QPCR measurements. Significance levels obtained from one-way ANOVA with Tukey's post-hoc test for pairwise comparison. Significance levels were denoted by *($p < 0.05$), **($p < 0.01$), ***($p < 0.001$), and ****($p < 0.0001$).

significantly across cell types (Fig. 4, see Supplementary Data 3). For ease of visualization of a multivariate dataset, hierarchical clustering was employed to group morphome features of high similarity together and thus reveal morphological profiles across different nanotopographies. Before clustering, each morphome feature was mean centered and normalized, transforming each morphome feature to a relative scale with negative values denoting

decrease and positive values denoting increase from the mean = 0. When taken entirely, hierarchical clustering of the morphome revealed distinct morphological profiles of all combinations of cell type and nanotopography (see Supplementary Fig. 7). Our results were further distilled to hierarchical clustering of the morphome separated by cell type (Fig. 4 and Supplementary Data 4). When compared to FLAT, pre-myoblasts on SQ showed high average

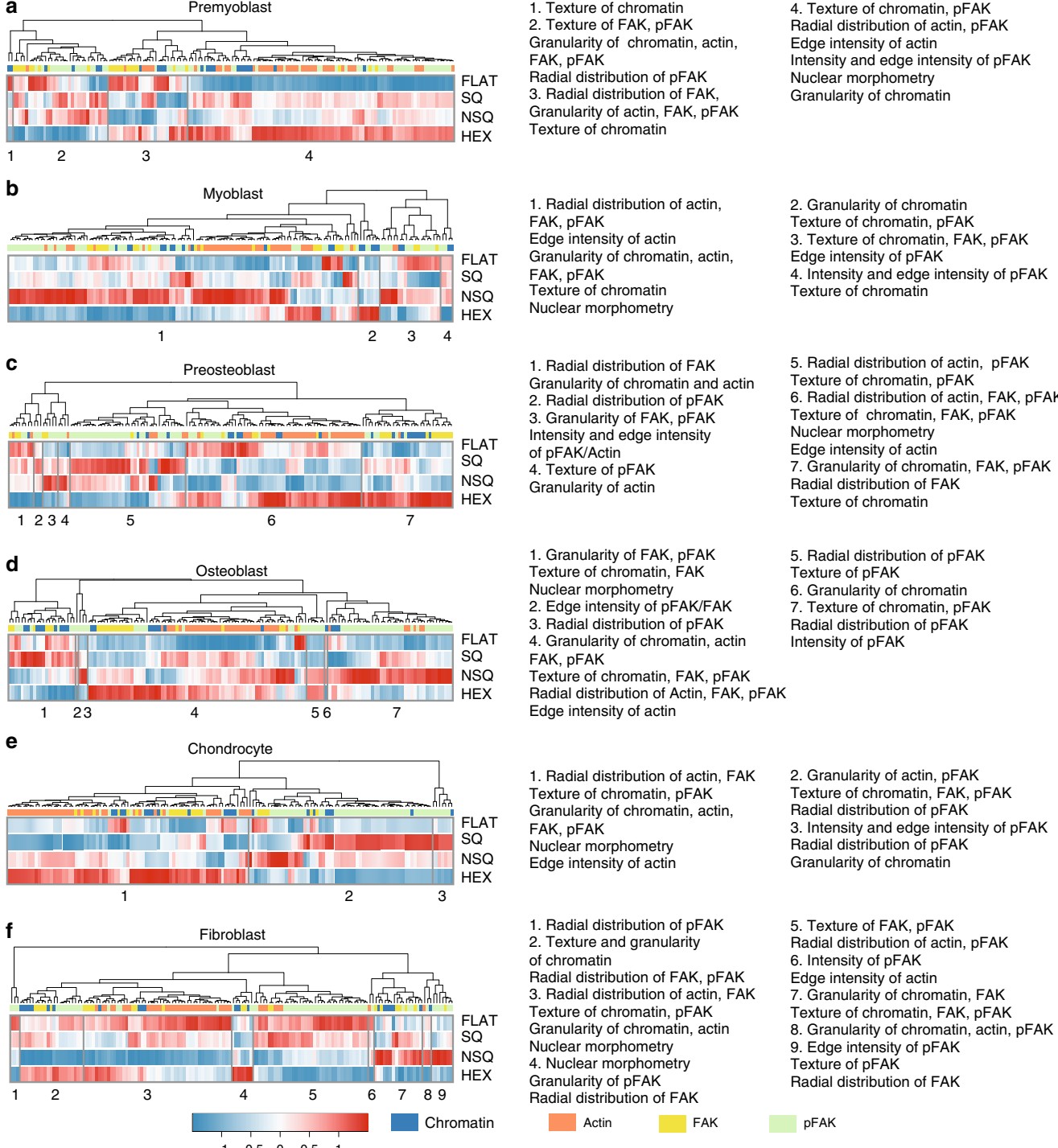

**Fig. 4 The morphome reveals nanotopography-specific changes to cell morphology.** Hierarchical clustering of the morphome within each cell type. The data morphome from each cell type were clustered separately (see Supplementary Data 5). Within the cell type-specific morphome, each morphome feature was normalized to have a mean = 0 and standard deviation = 1. The color and intensity of each tile represents the average value of the feature. The morphome features analyzed were comprises 21 chromatin, 42 actin, 23 FAK, and 60 pFAK features. Features included were changed significantly across topographies ($p < 0.05$ using one-way ANOVA). The total number of cells analyzed from two independent experiments were: **a** $n = 877$ pre-myoblasts; **b** $n = 931$ myoblasts; **c** $n = 644$ pre-osteoblasts; **d** $n = 728$ osteoblasts; **e** $n = 619$ chondrocytes; and **f** $n = 1140$ fibroblasts.

values of: focal adhesion textures, pFAK radial distribution, nuclear morphometry, and chromatin textures (Fig. 4a). The morphome of pre-myoblasts cultured on SQ reflects the need for FAK phosphorylation and preferential localization at stress fiber edges, which is necessary for myotube differentiation[32,33]. In contrast, myoblasts on SQ showed a particularly high average value for chromatin granularity and nuclear morphometry, and near-zero values for radial distribution of actin and of focal adhesions (Fig. 4b). High-chromatin granularity observed for both pre-myoblasts and myoblasts on SQ denotes chromatin hetero-geneity and condensation and transcriptional activity, which is reportedly higher prior to myotube formation[34,35].

Pre-osteoblasts on SQ and NSQ had high average values for pFAK radial distribution, intensity, granularity and texture, and high average values for granularity of chromatin and actin (Fig. 4c). However, pre-osteoblasts on SQ had higher order pFAK and FAK radial distribution than on NSQ, which induced the highest expression of osteogenic markers. The morphome of pre-osteoblasts grown on NSQ featured radially variable actin that resemble bone cells, which have high contractility and actin stress fibers[36].

For osteoblasts, the differences between the SQ and NSQ morphome were more prominent: NSQ induced lower average values of focal adhesion granularity, chromatin texture and nuclear morphometry, and higher average values for focal adhesion radial distribution (Fig. 4d) compared to SQ. The osteoblast morphome on NSQ indicates that focal adhesions localize at regular intervals along the periphery, which is associated with osteogenesis[37]. Furthermore, changes in nuclear morphometry attributed to spreading after growth on stiff surfaces is also associated with osteogenic differentiation[38].

Chondrocytes on HEX, which significantly increased chondrogenic marker gene expression relative to FLAT, showed high average values of radial distribution, texture and granularity of actin and FAK, high average nuclear morphometry, and low average values of pFAK and chromatin measurements (Fig. 4e). These characteristics reflect the morphological changes (including reduced contractility and stress fiber formation, increased cell circularity, and decreased cell spreading[36], low-FAK phosphorylation[39] and poor focal adhesion formation[40]) of stem cells undergoing chondrogenesis.

The morphome of fibroblasts cultured on FLAT had high average values of both actin and focal adhesion measurements (Fig. 4f). The highly uniform radial arrangement of focal adhesions and actin of cells on FLAT indicate reduced polarization and contractile morphology of fibroblasts activated to a fibrotic state[41]. Inflammation pathways are reportedly increased in fibroblasts on HEX[42], inducing low adhesion that is reflected in low actin and focal adhesion radial distribution. Fibroblasts grown on NSQ and HEX showed low average values of focal adhesion and actin radial distribution but high values when grown on SQ.

Overall, the morphome reflected cell-type-specific responses to nanotopography. This was highlighted by the dissimilarity of the hierarchically clustered morphome from cell types with similar lineage or origin (e.g., pre-osteoblasts vs. osteoblasts, Table 1). Furthermore, by training a Bayesian logistic regression classifier using the morphome as predictors we confirmed that the morphome contains sufficient information to distinguish six different cell types. We observed robust classification of cell type using either an out-of-sample morphome or a morphome obtained from the same dataset (see Supplementary Tables 3 and 4, and Supplementary Note 2), demonstrating the fact that the morphome contains sufficient information to describe cell types. The logistic regression classifier also indicates the radial arrangement of actin and focal adhesions were critically distinct between musculoskeletal cell types, while the arrangement of actin fibers into stress fibers or into cortical, circular bundles provided information on various cell states.

We also clustered the morphome based on nanotopography (see Supplementary Fig. 8 and Supplementary Note 3). Patterns emerged in the morphome in direct response to nanotopography: NSQ induced high average values of pFAK radial distribution, texture and granularity; and HEX induced high average values of actin radial distribution. Correlation analysis between the dendrograms confirm that the morphome clusters of different nanotopographies were dissimilar to each other (see Supplementary Table 5).

**The morphome robustly predicts gene expression**. The Spearman rank correlation revealed that varying degrees of correlation exist between morphome features and gene expression (see Supplementary Fig. 9). We hypothesized that the morphome would sufficiently encompass cell response induced by nanotopography. Thus, we utilized Bayesian linear regression to predict gene expression using the morphome features as predictors (for the explicit model definition, see Methods). A Bayesian linear regression model reflects uncertainty in the estimation of regression weights compared to point value estimates using maximum likelihood regression. Gene expression was modeled independently of each other, thereby creating 14 different equations with variable weighting of the morphome features. Importantly, the model was trained without any prior knowledge of cell type and topography type or parameters (e.g., nanodot diameter or center-to-center distance), instead relying on the morphome to encode both information.

The morphome clearly captured gene expression changes induced by nanotopography (Fig. 5a). The heterogeneity inherent in single cells, usually uncaptured by population measurements of gene expression, are apparent in the variance of the predictions using the model. The mean absolute error (MAE) for prediction of all genes was between 10% (for prediction of *MYOD1*, *MYOG*, and *MYH7*) and 21% (for prediction of *COL3A1*, see Supplementary Table 6).

The magnitude of the regression weight reflects the contribution of each morphome feature in predicting gene expression (see Supplementary Data 5). Across all 14 genes, pFAK activation, as indicated by pFAK/FAK integrated intensity ratio, consistently contributed to the prediction of all 14 genes. FAK texture and radial distribution, actin texture, and chromatin granularity features considerably contributed to prediction of gene expression (see Supplementary Fig. 10). pFAK was particularly important to the model due to its relevance in contractility induced by nanotopography[10], fibrosis and scar tissue formation[43], in vitro osteogenesis[44,45], and chondrogenic maintenance[39].

**Table 1 Correlation coefficient of morphome features hierarchically clustered by cell type.**

|  | All cell types | Pre-myoblast | Myoblast | Pre-osteoblast | Osteoblast | Chondrocyte | Fibroblast |
|---|---|---|---|---|---|---|---|
| All cell types | 1 |  |  |  |  |  |  |
| Pre-myoblast | 0.392 | 1 |  |  |  |  |  |
| Myoblast | 0.274 | 0.280 | 1 |  |  |  |  |
| Pre-osteoblast | 0.429 | 0.236 | 0.218 | 1 |  |  |  |
| Osteoblast | 0.410 | 0.201 | 0.280 | 0.354 | 1 |  |  |
| Chondrocyte | 0.540 | 0.164 | 0.374 | 0.393 | 0.397 | 1 |  |
| Fibroblast | 0.473 | 0.313 | 0.408 | 0.350 | 0.381 | 0.485 | 1 |

Higher correlation coefficient denotes higher similarity between groupings of morphome features between two dendrograms (resulting from hierarchical clustering of the morphome).

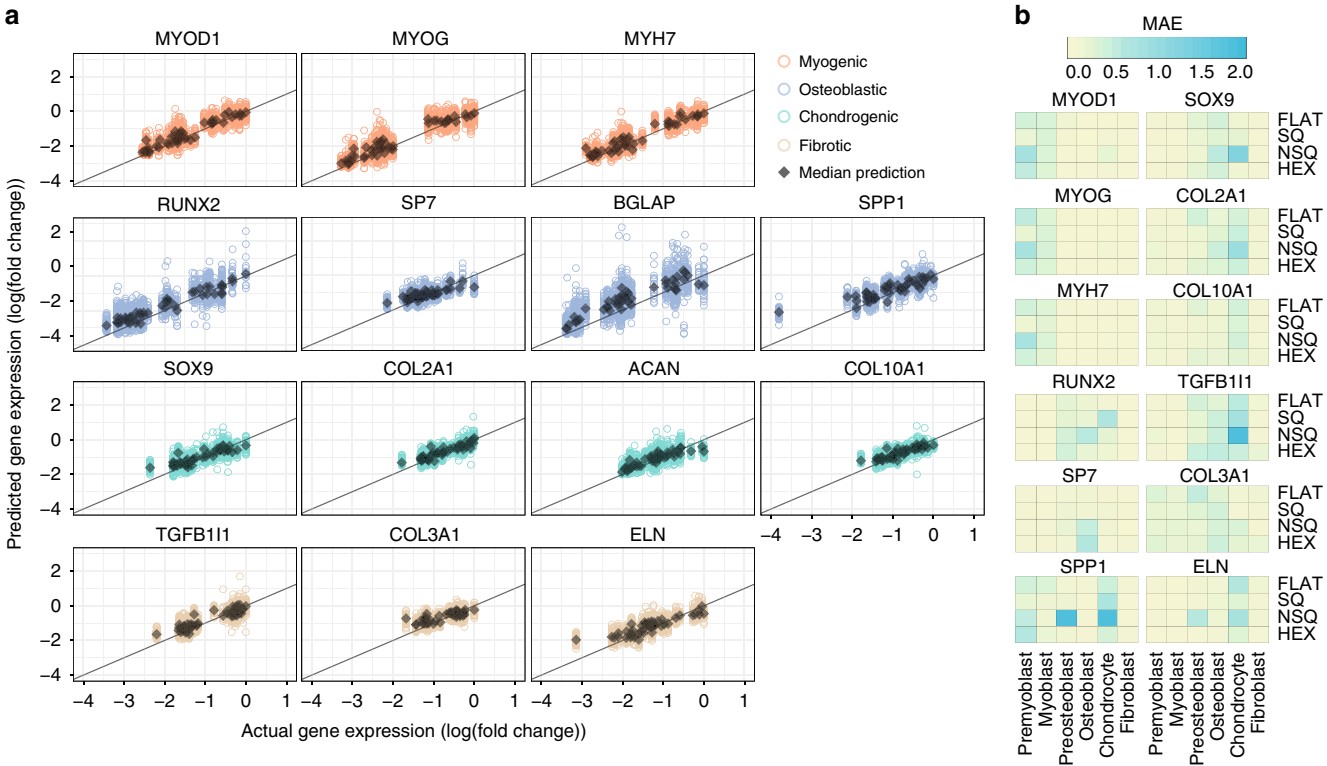

**Fig. 5 The morphome reliably predicts nanotopography-induced gene expression. a** The morphome was used to train a Bayesian linear regression model that predicted myogenic, osteogenic, chondrogenic, and fibrotic gene expression. The expression of each gene (response) was trained against linear combinations of morphome features (predictors) and without any prior knowledge on topography parameters. The linear regression model was trained using 60% of the dataset and tested using 40% of the data. Scatterplots show actual and predicted gene expression values by using the test set as input to the model, with each open faced circle showing predicted and actual gene expression from a single morphome. Black diamond shows the median predicted gene expression values. Colors represent the different musculoskeletal genes, with orange denoting myogenic, blue denoting osteoblastic, green denoting chondrogenic and brown denoting fibrotic genes. Mean absolute error (MAE) was obtained by first calculating the difference between actual and predicted gene expression values for each morphome then averaging differences across the entire morphome. **b** Testing the predictive power of the morphome by leave-one-out validation. To test the predictive power and bias of the morphome, the linear regression model was retrained after exclusion of one combination of cell type and topography. The excluded cell type and topography dataset was used for prediction, from which MAE was calculated. The tile position denotes the cell type and nanotopography combination that was excluded in the model and used for testing, while the color of each tile denotes the MAE.

The sensitivity and predictive power of the model was verified by iteratively training new models using a morphome with a held-out combination of cell type and topography (Fig. 5b). Essentially, we exploited the presence of multiple cell types and nanotopographies to be able to train predictive models of gene expression without overfitting to data from any experimental setup (e.g., background staining[46]). Drastic increases in MAE were observed when predicting lineage-specific genes using models that excluded the particular cell type lineage being tested, regardless of nanotopography. The results are logically explained by the fact that a particular cell type contributes the most information to gene expression prediction by virtue of its lineage. Removal of the morphome from the particular cell type in question thus drastically reduces the amount of distinct information in the model. Model prediction after removal of nanotopographies showed consistency in MAE, indicating the generalizability of the model in predicting gene expression from nanotopographies outside of FLAT, SQ, NSQ, and HEX.

**The morphome predicts cell behavior in a complex environment.** We demonstrate the application of the linear regression model by predicting gene expression of pre-osteoblasts and fibroblasts co-cultured on nanotopographies. A new morphome

was obtained from all cells on the entire nanotopography surface (see Supplementary Fig. 11). This co-culture morphome was then used as input in the model to predict gene expression (see Supplementary Fig. 12).

For visualization, the sum of predicted osteogenic (*RUNX2, SP7, BGLAP,* and *SPP1*) and fibrotic (*TGFB1I1, COL3,* and *ELN*) genes was plotted against the spatial coordinates of the pre-osteoblasts and fibroblasts. Osteogenic gene expression was highest on NSQ, wherein concentrated areas of enhanced expression levels were present (Fig. 6a). These areas might represent hotspots or nuclei of osteogenic paracrine signaling induced by the NSQ nanotopography[8]. In contrast, osteogenic gene expression was low and homogenous on the FLAT, SQ, and HEX topographies. The uniformity of cell distribution across each nanotopography (see Supplementary Fig. 11) eliminates the possibility of inadvertent cell clustering as the origin of gene expression changes.

Fibrotic gene expression showed more spatial variability across nanotopographies but was also maximized on the NSQ nanotopography, and largely overlaps with the spatial pattern of osteogenic gene expression (Fig. 6b). This is attributable to the synergistic interaction of osteoblasts and fibroblasts on osteogenic differentiation and mineralization[47]. The predicted effect of high osteogenic gene expression induced by NSQ was verified in the

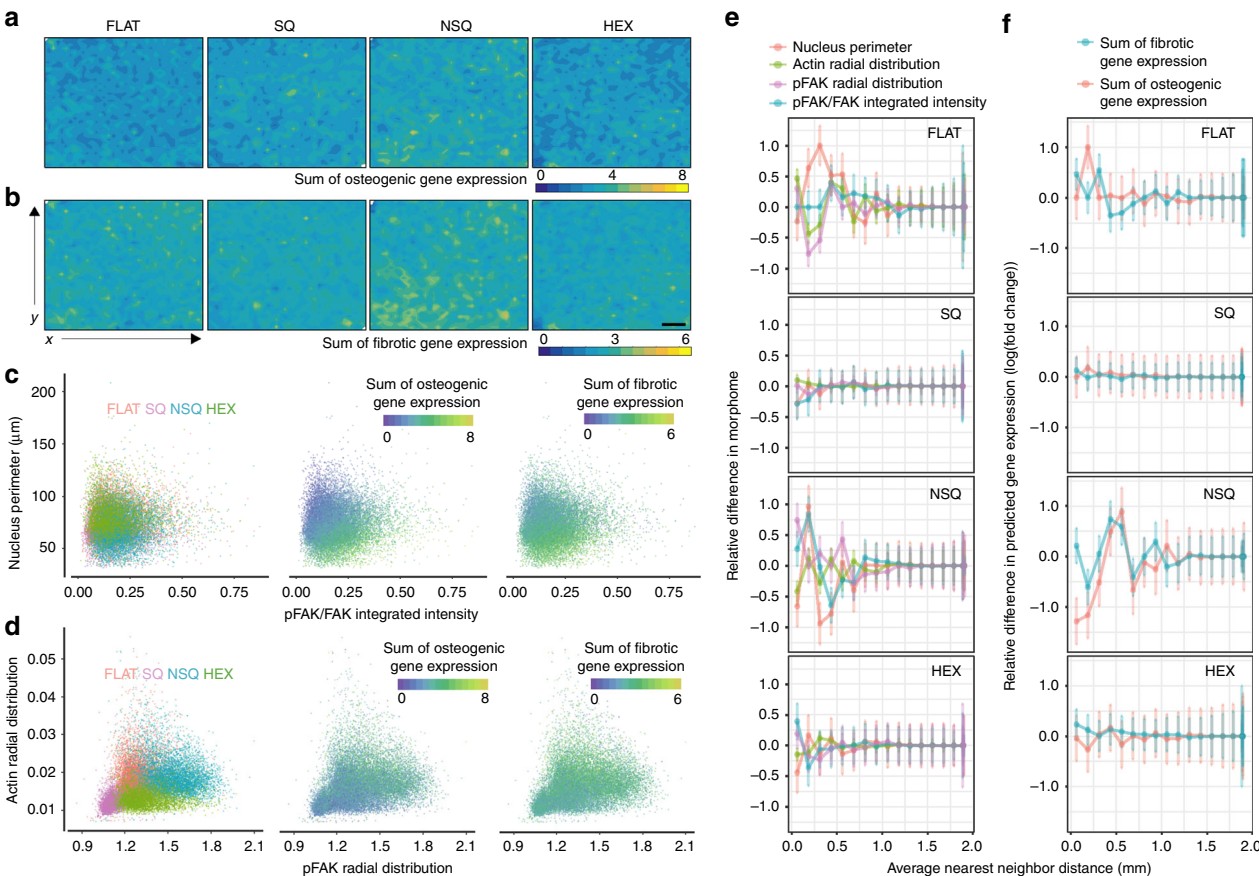

**Fig. 6 Single-cell analysis of morphology and gene expression using the morphome. a, b** Predicted response of a pre-osteoblast and fibroblast co-culture to nanotopography. Contour plots show the sum of predicted **a** osteogenic (*RUNX2, SP7, BGLAP, SPP1*) and **b** fibrotic (*TGFB1I1, COL3, ELN*) gene expression for individual cells on FLAT, SQ, NSQ, and HEX topographies. Pre-osteoblast and fibroblast cells were co-cultured on FLAT, SQ, NSQ, and HEX nanotopographies for 2 days, and their morphome obtained from the entire nanotopography surface. The newly collected morphome was then used as input in the linear regression model (shown in Fig. 5a) to predict gene expression. *x* and *y* axes of each contour plot shows are spatial coordinates on the nanotopogrpahy substrate, while the color of the contour represents the level of summed gene expression. Scale bar = 100 μm. **c, d** Morphology and gene expression at the single-cell level is provided by the morphome. Each dot in the scatterplot denotes a single-cell. Nanotopographies are color coded, with FLAT denoted in pink, SQ denoted in purple, NSQ denoted in blue and HEX denoted in green. **e, f** Cell–cell interaction altered by nanotopography. The average changes in **e** cell morphology and **f** gene expression between two neighboring cells separated by a specified distance was measured and normalized to the maximum observed change. Data are presented as mean ± standard deviation and reported as a function of distance between two cells binned every 125 μm. N ≥ 5000 cells per topography from one independent experiment.

increased osteogenic gene expression at 7 days and mineralization compared to FLAT at 28 days (see Supplementary Fig. 13). Given that this model is based on single-cell populations, our predictions using the co-culture morphome indicate the capability to encode not just cell-material interaction but a concerted response arising from the cellular milieu.

We additionally gleaned insights in single-cell responses induced by nanotopography using the morphome. The combination of spatial information and cellular response to nanotopography present in the morphome permits analysis of single cells similar to flow cytometry. As an example, we focused on correlating morphome features with high contribution to the model. Nanotopographies induced indistinct effects on either nuclear perimeter or pFAK activation, yet there was a clear gradient in predicted osteogenic gene expression (Fig. 6c). As nuclei became smaller and pFAK activation increased, both osteogenic and fibrotic gene expression increased. In contrast, nanotopographies exhibited clearly separable effects on actin and pFAK radial distribution, with cells on SQ showing the lowest values (Fig. 6d). These particular changes in cell morphology correlated more with predicted osteogenic gene expression than

fibrotic gene expression, which lacked clear separation by nanotopography. From the morphome, we found strong evidence to implicate nucleus shape and pFAK activation changed by nanotopography as drivers of osteogenesis.

The morphome additionally uncovered effects of nanotopography on cell–cell interaction at the morphological level and on predicted gene expression (Fig. 6e, f). On average, the effect of FLAT on cell–cell interaction and the resulting cell morphology extended up to 1 mm (Fig. 6e), yet gene expression changed maximally at a separation distance of only 250–375 μm between neighboring cells (Fig. 6f). In contrast, the average effect of SQ and HEX on cell–cell interaction and morphology were minute and apparent only at short cell–cell separation distances of 250 and 500 μm, respectively. A predominantly negative effect on pFAK activation was observed between neighboring cells grown on either SQ or HEX. The suppressive effect of nanotopography on cell–cell interaction correlate strongly with the homogeneous gene expression predicted for SQ and HEX (Fig. 6f). On the contrary, long-range effects between cells on NSQ were distinct (Fig. 6e). In contrast to FLAT, neighboring cells on NSQ separated by 1 mm or less selectively exhibited drastic changes

only in nucleus perimeter and pFAK activation. The long-range effects of cell–cell interaction observed in NSQ were clearly manifested in predicted gene expression (Fig. 6f). In fact, NSQ showed a critical distance of 500–625 μm between neighboring cells where fibrotic and osteogenic gene expression were maximally changed. In summary, we observed a clear augmentation in cell–cell interaction induced by NSQ compared to FLAT, SQ and HEX. This long-range interaction between neighboring cells on NSQ separated by 1 mm or less drove changes in pFAK activation and osteogenesis.

## Discussion

In this study, we present a system that robustly uses morphology to quantitatively predict cell type-specific responses to nanotopography. The morphome, which in this study is the collective morphological measurements of chromatin, actin and focal adhesions within single cells, were found to manifest nanotopography-induced changes in cell morphology and gene expression. The information in the morphome underpinned the robustness of a Bayesian linear regression model for predicting gene expression. The morphome also uncovered biological insights at both the morphological and gene expression levels resulting from nanotopographical perturbation of a complex cell microenvironment.

The morphome-based predictive model reported here offers two distinct advantages over the current state of the art. First, our predictive model utilizes hundreds of cell morphology features accurately predict expression levels of 14 different genes. Moreover, because our model exploits the relationship of the morphome to cell type-specific gene expression, it can be used to predict gene expression induced by new microenvironments, given a set of images of cells grown on them. Some groups have previously used cell morphology to glean insights on cell lineage commitment[36], cell response to topography[20,21,48], and design rules controlling cell behavior[20,21,48]. However, all of these studies only go as far as describing correlations between the cell microenvironment and morphology from a single-cell type for classification.

Second, we use gene expression to determine magnitude of the cell response to nanotopography. This contrasts with the current state of the art[48], relying heavily on setting arbitrary boundaries in protein or gene expression levels to classify cells into functional cell classes. While this work does not indicate that our nanotopographies maximize functionality at 7 days, our predictive model allows us to easily rank nanotopographies in their effect on different musculoskeletal lineages. An important consequence of this work is in utilizing predicted gene expression levels to compare new topographies. Indeed, our morphome-based approach supports a function-focused exploration of new topographies that will make the current trial-and-error screening approach more efficient. Moreover, since gene expression is highly scalable, our system can be easily adapted to predict the expression level of any gene of interest.

The utility of our morphome-based models was validated by robust prediction of the outcome from a co-culture of osteoblasts and fibroblasts. The complexity of a co-cultured microenvironment prevents direct inference from our single-cell culture results. Yet the osteogenic function induced by NSQ from a complex co-culture system was predicted by the morphome models, and validated by high mineralization observed after 28 days of culture. Our results suggest that the morphome can also manifest cellular changes driven by chemical or paracrine cues. This property of the morphome can be exploited to predict cell behavior in more complex microenvironmental settings.

The co-culture experiment also showed that the morphome dataset encompasses, at high resolution, structural, functional and spatial information. Indeed, we took advantage of this informationally rich dataset to uncover enhancement of cell–cell interaction (from micron to millimeter range) resulting from a subtle change in nanotopography order. SQ and HEX, both of which present an ordered interface to the cell, suppressed cell–cell interaction while cell–cell interaction was apparent at long distances on FLAT and NSQ. This result presents a new mechanism for nanotopography-induced-cell behavior.

Clearly, morphome capture is crucial to the ability of the linear regression model to predict nanotopography-induced gene expression. While population-level measures of gene expression strongly indicate cell function, they introduce a measure of uncertainty and biological variability into the linear regression model. Thus, a one-to-one relationship between the morphome and cell function is essential to develop. Non-destructive microscopic and molecular tools[49] that combine spatial and structural information from the morphome with single-cell functional assays are vitally important for establishing quantitative topography structure- and cell-function relationships using the morphome. However, the use of routine methods, such as high-content imaging and QPCR, permits any lab to measure the morphome and to model it against the gene expression in question.

By generating a multivariate morphome dataset and combining it with machine learning, we have created a powerful platform for relating topography structure to gene expression. The predictive power of the Bayesian linear regression model presented here easily lends to sequential experimental design by exploiting uncertainty and variability within the model[50]. Combined with bench-top lithographic techniques[51] and in silico simulation of morphological response to nanotopography[52], we envision a completely closed-loop system that enables functionally oriented exploration of new topographies.

## Methods

**Polycarbonate surfaces with nanotopography.** Surfaces patterned with 120 nm diameter and 100 nm depth nanopits were fabricated on polycarbonate using injection molding[53]. The following nanotopographies were used: surfaces without nanopits (FLAT); nanopits in a square array with 300 nm center-to-center spacing (SQ); nanopits in a square array with ~300 nm center-to-center spacing distorted by 50 nm in both $x$ and $y$ directions (NSQ); nanopits in a hexagonal array with 300 nm center-to-center spacing (HEX). Samples were cleaned in 70% ethanol and dried before treating with $O_2$ plasma at 120 W for 1.5 min. Samples were sterilized using UV light in a biological safety cabinet for at least 20 min before cell seeding.

**Cell culture.** Mouse fibroblast cell line NIH3T3 (ATCC) was cultured in reduced sodium bicarbonate content (1.5 g per liter) Dulbecco's modified Eagle's medium with (DMEM) supplemented with L-glutamate (2 mM), 10% bovine calf serum, and 1% penicillin–streptomycin. Mouse C2C12 myoblasts (ATCC) were cultured in DMEM with 20% FBS and 1% penicillin–streptomycin, and committed into mature myoblastic cells using DMEM supplemented with 2% horse serum and 1% penicillin–streptomycin[32,33]. Mouse chondrocytes were cultured in minimum essential medium alpha (MEMα) with nucleosides, ascorbic acid, glutamate, sodium pyruvate supplemented with 10% FBS and 1% penicillin–streptomycin. Mouse MC3T3 cells (ATCC) were cultured in MEMα with nucleosides and L-glutamine without ascorbic acid and supplemented with 10% FBS and 1% penicillin–streptomycin. To commit MC3T3 into mature osteoblasts, MC3T3 media was supplemented with 10 nM dexamethasone, 50 μg per ml ascorbic acid and 10 mM β-glycerophosphate[27,54]. Lineage committed progenitor cells, referred here as pre-osteoblasts and pre-myoblasts, were also included in the study to mimic the osteogenic and myogenic regeneration profile in the adult tissue[27,28].

**Cell seeding.** Cells were harvested from flasks using trypsin in versene buffer and spun down at $400 \times g$ for 5 min. NIH3T3 and MC3T3 cells were resuspended in complete media and seeded at 4000 cells per $cm^2$. Chondrocytes and C2C12 were seeded at 2500 cells per $cm^2$. Cells were seeded at different densities to ensure single cells at ~30% confluency on each surface after 2 days culture. To ensure homogeneity of seeding, cells were seeded using a device that controls fluid flow[55].

For co-culture studies, MC3T3 and NIH3T3 cells were simultaneously seeded at 2000 cells per cm$^2$ per cell type in MC3T3 growth media. All cells were grown on nanotopographies for either 2 days (for image-based cell profiling) or 7 days (for gene expression measurement).

**Gene expression measurement**. After 7 days, total RNA was obtained from lysed cells according to manufacturer's instructions (Promega ReliaPrep Cell Miniprep kit). Gene expression was measured directly from 5 ng RNA using a one-step QPCR kit with SYBR dye (PrimerDesign). A list of the forward and reverse primers used to study different mouse genes is provided in Supplementary Table 7. QPCR was run on the BioRad CFX96 platform. Relative gene expression was normalized to the 18S ribosomal RNA reference gene. Gene expression was measured at least twice from each independent experiment. One-way analysis of variance (ANOVA) with Tukey's post-hoc test for multiple comparisons was performed to determine the effect of nanotopography on gene expression compared with FLAT. Statistical significance was considered at $p < 0.05$. Plotting and statistical testing of gene expression data were performed using GraphPad Prism (v7.0a).

**Immunofluorescence staining**. After 2 days, cells on surfaces were fixed with 4% paraformaldehyde solution in phosphate buffered saline at 4 °C for 15 min. Fixed cells were then permeabilized and blocked with 10% goat serum and 2% bovine serum albumin in phosphate buffered saline for 1 h at room temperature. Cells were stained with the following primary antibodies overnight at 4 °C: pFAK Y397 (Abcam 39967, 1:400) and FAK (ThermoScientific 396500, 1:400). Afterwards, Alexa Fluor-conjugated secondary antibodies (ThermoScientific, 1:500) against the host species of the primary antibody were used. Alexa Fluor 549-conjugated phalloidin (ThermoScientific, 1:200) were used to visualize the actin cytoskeleton. Cells were also stained with 4′,6-diamidino-2-phenylindole (DAPI; (Thermo-Scientific) to visualize chromatin inside the nuclei. DAPI was previously reported to contribute textural information as a means of alternatively representing chromatin[56]. All surfaces were mounted on 0.17 µm thick glass coverslips with ProLong mounting medium (ThermoScientific) and dried overnight at 4 °C before imaging.

**Image acquisition and morphome extraction**. For single population studies, monochrome images of each fluorophore were obtained at x40 magnification (numeric aperture 1.3) using the EVOS FL1 System (ThermoScientific). For co-culture studies, the entire nanotopography surface was imaged and stitched through an automated microscope (EVOS FL2 Auto) with a x40 magnification (numeric aperture 1.3). All images from the same cell type were obtained using the same camera and light settings. Afterwards, image processing and morphome feature extraction were perfomed using CellProfiler[57] (v2.4.0, The Broad Institute). Image processing, including illumination correction and channel alignment, was performed across each independent experiment from the same cell type[58]. Nuclei and cells were segmented from each image, allowing single-cell analysis of shape or morphometric measurements, total and local intensities and textures from chromatin, actin, pFAK and FAK. Measurements were taken from distinct cells from each independent experiment.

**Multivariate analysis**. The morphome initially consisted of a total of 1050 measurements obtained from single cells. Features with zero variance within each batch (e.g., Zernike Phase measurements) were first removed from the dataset. Morphome measurements from single-cell populations and independent experiments were first combined then scaled by subtracting the mean and normalizing by the standard deviation of the dataset to result in a dataset with mean = 0 and standard deviation = 1. Morphome data from co-culture studies were similarly scaled and normalized using the mean and standard deviation from the initial dataset consisting of single-cell populations. A Pearson correlation method at significance level 90% was used to remove features with correlation higher than 0.9 without significantly reducing total data variance (see Supplementary Fig. 14) using the KMDA (v1.0) package for R[59]. After pre-processing, 624 morphome features were used in the study.

**Hierarchical clustering**. To determine the features that were significantly varied across nanotopography, a one-way ANOVA with Tukey's post-hoc test for multiple comparisons was performed. Prior to clustering, each morphome feature was mean centered and normalized to the standard deviation. An agglomerative hierarchical clustering algorithm was performed using a Euclidean distance metric and an average linkage method for cluster linkage using gplots (v3.0.1) package[60]. Membership of each morphome in a cluster was obtained from silhouette analysis using the cluster (v2.0.7-1) package[61]. Hierarchical clustering was visually represented with a dendrogram, and a heatmap with color intensity corresponding to the average values of the morphome features. Dendrogram correlation, which measures the similarities in the grouping of morphome features between two dendrograms, was performed using the corrplot package[62].

**Bayesian linear log-Normal regression**. Only morphome features with an absolute Spearman correlation coefficient ≥0.7 against all examined gene

expression markers were used in the linear regression model. The linear regression model used 243 morphome features, containing 22 nuclear morphometry and chromatin, 71 actin, 75 FAK and 75 pFAK measurements, as predictors of the model. QPCR data from each independent experiment was propagated across the corresponding single-cell morphome data from the same independent experiment. For each gene analyzed, data were rescaled from 0 to 1 by normalizing to the maximum gene expression.

Linear regression was performed as a simple approximation of the relationship between the morphome and myogenic, osteogenic, chondrogenic, and fibrotic gene expression. Established Bayesian inference methods were used to determine the probabilities of observing gene expression with a given morphome set. We consider a linear model where expression of one gene (response y) was predicted through a linear combination of the morphome features (predictors X) transformed by the inverse identity link function. We assume that y follows a log-Normal distribution parametrized by the mean $\mu$ and variance $\sigma_i^2$:

$$y_i \sim \log \mathrm{Normal}\left(\mu_i, \sigma_i^2\right) \qquad (1)$$

And that $\mu$ is a linear function of X parametrized by $\beta$:

$$\mu_i = \beta_0 + \beta_1 X_1 + \beta_2 X_2 + \ldots + \beta_n X_n \qquad (2)$$

All model parameters $\beta$ were assumed a priori to come from a normal distribution, parametrized by mean and standard deviation:

$$\beta \sim \mathrm{Normal}(0, 2) \qquad (3)$$

Each gene was trained independently resulting in 14 different linear regression equations. A 60–40% training and test split for Bayesian linear regression was performed randomly and with stratification using the caret (v6.0-81) package for R[63]. The Bayesian linear model was created using the brms (v2.5.0) package for R[64], which utilizes the Hamiltonian Markov Chain Monte Carlo sampler for estimation of the posterior distribution of $\beta$. Bayesian linear modeling was carried out using with 1000 warm-up iterations and 1000 sampling iterations within each chain for three independent chains. All models were confirmed to converge to the equilibrium distribution by confirming low-autocorrelation, potential scale reduction statistic split $R_{\mathrm{hat}} \geq 1$, effective sample size was smaller than total sample size. We confirmed the suitability of the prior distribution by ensuring that the data regenerated using the prior predictive distribution (i.e., without seeing any data) closely aligned with the real dataset. Predicted gene expression was performed by using the test set or the morphome obtained from the co-culture study as input to the linear model. Predicted values were averaged across 50 draws from the posterior distribution. The magnitude of the average values of parameters $\beta$ indicated feature importance as it effectively weighted the contribution of each morphome feature in predicting gene expression.

To determine the predictive power of the morphome, a specific combination of cell type, topography and independent experiment or replicate were iteratively omitted, and the remaining dataset was used to refit new models. Thus, 576 additional models were created to test 24 different cell type combinations across 12 genes and 2 independent experiments. The predictive quality of the models was assessed by predicting the expression of all 14 genes from the held-out cell type, topography and replicate dataset. We report the mean absolute error (MAE) of QPCR prediction for each cell type and topography combination averaged across two independent experiments. MAE was calculated as the average across all absolute differences between predicted and actual gene expression.

**Analysis of cell–cell interaction**. The co-culture morphome was used to predict gene expression at the single-cell level. The dataset was then used to determine changes in gene expression and morphome between neighboring cells of a given distance. Changes in cell morphology and gene expression was performed for each cell against all other cells. Distances between cells were binned to calculate average change in cell morphology and gene expression at intervals of 125 µm. The changes in cell behavior between two cells was normalized to the maximum value of change observed.

**Statistics, visualization, and software**. Statistical analysis and machine learning were performed using statistical software R (v3.4.3) and its graphical interface RStudio (v1.0). Scatterplots, boxplots, and histograms were generated using ggplot2 (v3.1.0) in R[65]. Interpolation of x and y coordinates for contour plotting was performed using bivariate interpolation of a regularly gridded dataset using akima (v0.6-2) package[66]. Contour plots were created using the fitted.contour function in R, with the nuclear centroid position used as spatial coordinates of the cell. Barcharts and one-way ANOVA analysis of QPCR values were obtained using GraphPad Prism (v7.0a).

**Reporting summary**. Further information on research design is available in the Nature Research Reporting Summary linked to this article.

## Data availability

Raw data (e.g., gene expression data, morphome data), R workspace data that contains all Bayesian linear regression models, and associated code that support the findings of this study are available in Zenodo with the identifier 10.5281/zenodo.3608197

## Code availability

The statistical models proposed and evaluated in this paper is realized using standard packages in R. The code used to create models and R workspace containing all fitted models are available in the dataset published on Zenodo with the identifier 10.5281/zenodo.3608197

## Material availability

Nanotopographies used in this study are made in house and can be obtained from the corresponding author upon reasonable request.

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

## Acknowledgements

We acknowledge ERC funding through FAKIR 648892 Consolidator Award. MFAC is financially supported by the University of Glasgow MG Dunlop Bequest, College of Science and Engineering Scholarship, and FAKIR consolidator award. NG acknowledges support by the Research Council of Norway through its Centres of Excellence funding scheme, project number 262613. We acknowledge the James Watt Nanofabrication Center for fabrication work, and Steen Lillelund for initiating the machine learning work. We thank Julie Russell for her contribution to the QPCR, Carmen Huesa for providing the primary cartilage cells and Rachel Love for the injection molding of nanotopographies.

## Author contributions

M.F.A.C., P.M.R., and N.G. designed the biological experiments. M.F.A.C. and B.S.J. designed the machine learning analysis. M.F.A.C. carried out imaging, image analysis, QPCR, machine learning. P.M.R. fabricated and characterized the nanotopographical surfaces. M.F.A.C., B.S.J., P.M.R., and N.G. wrote the manuscript. All authors have read and approved the manuscript before submission.

## Competing interests

The authors declare no competing interests.
