## [Peer Review File · Nature Communications]

Reviewers' comments:

Reviewer #1 (Remarks to the Author):

Most of my concerns have been addressed. There are 3 remaining:

1) It seems from the response to query 4 that the authors may not understand the concern, which is a serious one. It is true as they say that **[REDACTED]**

However, this is true *if and only if* every individual sample is completely experimentally independent, which is rarely the case. If any of the tested images in the cross validation come from the same sample, or the same experimental batch, then they are not independent and the potential for contamination exists. Here is a particular example where, by cross validation, everything appears to be fine but blacking out the cells in the images still enables high classification accuracy: <https://onlinelibrary.wiley.com/doi/abs/10.1111/j.1365-2818.2011.03502.x> This is because the classifiers were trained and tested on individual images from the same batch and the classifier learned to tell the difference between different background staining features in the images, rather than the morphology of the cells. This is why it is important to split the test/training so that no experimental batches are present in both.

2) The authors state: **[REDACTED]**

concern is the same as before: the use of **[REDACTED]** here is improper. **[REDACTED]** Again, my

"X" means that all other possible ways of X have been tested and morphome was the only successful one, which is not the case here. **[REDACTED]** is also an imprecise term to use in this context without evidence that it is literally comprehensive. The sentence in the abstract should therefore be changed. "We demonstrate that nanotopography-induced changes in cell phenotype (both morphological and functional) are uniquely encoded by the morphome." And phrases like "uniquely determines" should be changed to "distinguishes" or similar. I think it's just a misunderstanding of the word 'unique'.

3) I disagree with the author response to Reviewer 2 query 3, which was:

[REDACTED]

The authors reply

[REDACTED]

This is not

my primary research area, but this paper seems to falsify that statement:
<https://www.frontiersin.org/articles/10.3389/fbioe.2018.00087/full>

Reviewer #2 (Remarks to the Author):

The authors have done a satisfactory job responding to queries from the first review, including some new data and analysis. I recommend the paper be published as is.

Reviewer #3 (Remarks to the Author):

This paper from the Gadegaard lab as changed in a few ways, though I will highlight my extensive and on-going concerns that preclude my ability to support publication at this point. That said at the outset of my comments, I would like to thank the authors for clarification of my concerns with Figure 1.

1. I appreciate some of the language changes with Figure 2, but my two fundamental concerns remain. First, small changes in gene programs are called significant when those subtle changes may not amount to a major push towards one lineage; myogenic and chondrogenic markers are especially uninspiring. Second, there is no positive control. Without it, how good can we say the HEX pattern or any other is? If your pattern induces minor changes, does that really mean the cells have become X, Y, or Z? Not likely even if many markers show that. There are also no comments on whether the cells are recapitulating developmental pathways.

My concerns were met with the statement that a **[REDACTED]** of the exercise. I got back to my statement above: I don't care about small but robust changes. I want something that will recapitulate the appropriate developmental program. The authors have not provided any response, text changes, or new experiments to assuage my concerns.

2. My simple statement that the authors should comment on variance between cells was taken out of context stating that **[REDACTED]**

While I agree that FLAT can show the net change from baseline, it does not describe the variance in response between donors or between lines of the same cell type.

I appreciate the changes that were made but these remain concerns, especially for a more general audience journal as Nat Comm.

Predicting gene expression using morphological cell responses to nanotopography

Marie F.A. Cutiongco¹, Bjørn S. Jensen², Paul M. Reynolds¹, Nikolaj Gadegaard^{1,*}

¹Division of Biomedical Engineering, School of Engineering, University of Glasgow, Glasgow, United Kingdom

²School of Computing Science, University of Glasgow, Glasgow, United Kingdom

* corresponding author

Our responses to reviewers are in blue, with accompanying changes in the manuscript highlighted in yellow.

Reviewer # 1:

1. It seems from the response to query 4 that the authors may not understand the concern, which is a serious one. It is true as they say that

[REDACTED]

However, this is true *if and only if* every individual sample is completely experimentally independent, which is rarely the case. If any of the tested images in the cross validation come from the same sample, or the same experimental batch, then they are not independent and the potential for contamination exists. Here is a particular example where, by cross validation, everything appears to be fine but blacking out the cells in the images still enables high classification accuracy:

<https://onlinelibrary.wiley.com/doi/abs/10.1111/j.1365-2818.2011.03502.x> This is because the classifiers were trained and tested on individual images from the same batch and the classifier learned to tell the difference between different background staining features in the images, rather than the morphology of the cells. This is why it is important to split the test/training so that no experimental batches are present in both.

We thank the reviewer for clarifying their concerns regarding the cross-validation method for classification. We have now altered our method of classifying the morphome into different cell types by using a rigorous leave-one-out scheme. This scheme is equivalent to what we presented in Figure 4B, which demonstrates the predictive capability of the morphome itself and without overfitting the data to experimental variation (e.g. in background staining, as described in the manuscript that this reviewer cited) . For this new classification scheme, we iteratively removed from the entire dataset a cell type from one independent experiment (the held-out dataset, e.g. Chondrocyte from the first independent experiment). The remaining dataset was used to train a Bayesian logistic regression model (details in Supplementary

Methods) to classify a cell type using the morphome. The left-out dataset was then used as test data to measure the accuracy of classification using the morphome. That is, in the training set we do not see examples of the cell types from a biological experiment, but we exploit the availability of other cell types to avoid overfitting to any individual experimental setup. We averaged the classification accuracy of each class from two independent experiments and present the average \pm standard deviation in Table S2 (also found below). In this manner, we obtained a measure of the predictive quality of the morphome that does not conflate information about the specific cell type in question from independent experiments. **Essentially this leave-one-out scheme allows us to predict outside of the dataset, an important task that supports our claim that the morphome contains sufficient information to classify cell types.**

Overall, we showed that classification of cells using the morphome still outperformed random classification (Table S2). This is similar to what we observed when we use the conventional machine learning method that separates the dataset into train and test sets without regard to experimental variation (Table S3). Both these results align with what we reported in Figure 4B. Through a similar leave-one-out scheme, Figure 4B emphasises how expression of genes related to a particular lineage suffers most from removal of the cell type committed to that lineage. This new information has been included in both the main manuscript and Supplementary Tables file.

Table S2. Accuracy of classifying the morphome into different cell types using Bayesian logistic regression with a leave-one-out scheme.

Bayesian logistic regression models were trained by holding out from the entire dataset a cell type from one independent experiment. The held-out dataset was used as test data to determine classification accuracy. We present here mean \pm standard deviation of classification accuracy from two independent experiments.

Cell type	Classification accuracy (%)
Pre-myoblast	50.6 \pm 7.6
Myoblast	52.4 \pm 10.1
Pre-osteoblast	63.4 \pm 4.7
Osteoblast	65.1 \pm 7.0

Chondrocyte	36.8 ± 4.3
Fibroblast	30.3 ± 7.8
Random	16.7

Table S3. Accuracy of classifying the morphome into different cell types using Bayesian logistic regression.

Bayesian logistic regression models were trained using a 60% of the entire dataset. The remaining 40% of the dataset was used to test accuracy of cell type classification.

Cell type	Classification accuracy (%)
Pre-myoblast	92.6
Myoblast	89.3
Pre-osteoblast	95.7
Osteoblast	95.2
Chondrocyte	97.2
Fibroblast	97.4
Random	16.7

2. The authors state:

[REDACTED]

Again,

my concern is the same as before: the use of **[REDACTED]** here is improper.

[REDACTED] "X" means that all other possible ways of X have been tested and morphome was the only successful one, which is not the case here. **[REDACTED]** is also an imprecise term to use in this context without evidence that it is literally comprehensive. The sentence in the abstract should therefore be changed. "We demonstrate that nanotopography-induced changes in cell phenotype (both morphological and functional) are uniquely encoded by the morphome." And phrases like "uniquely determines" should be changed to "distinguishes" or similar. I think it's just a misunderstanding of the word 'unique'.

We have now amended the manuscript to change the instances of "unique" or "entirely" or "uniquely determines".

3. I disagree with the author response to Reviewer 2 query 3, which was:

[REDACTED]

The authors reply

[REDACTED]

This is not my primary research area, but this paper seems to falsify that statement: <https://www.frontiersin.org/articles/10.3389/fbioe.2018.00087/full>

We appreciate the Reviewer's comment that perhaps comes from the lack of clarity in the contribution of our work relative to the current state-of-the-art. We have added the following paragraph to the discussion (page 23), stating how this study improves on what is currently known in the field, which we hope will expand the appeal of our work to non-experts.

Several groups have previously harnessed the informative content of the morphome to glean new insights on cell lineage commitment^{16,18,40}, cell response to topography^{15,17}, and design rules controlling cell behaviour^{15,17}. However, many of these studies (even more recently published ones^{58,59}) only go as far as describing correlations between the cell microenvironment, cell morphology and an arbitrarily specified functional cell category. In contrast, our work presents a predictive model that is multivariate in every sense: in its use of hundreds of cell morphology features to predict gene expression, and in its ability to predict expression levels of genes from 4 different cell functionalities. And from this multivariate approach, we are able to robustly predict actual, numerical quantities that define cell function. Indeed, another important consequence of this work is in allowing gene expression values to determine cell functionality stimulated by topography. This contrasts with the current state of the art, relying heavily on setting arbitrary boundaries in protein or gene expression levels to arbitrarily set cells into functional cell classes.

Reviewer #3:

1. I appreciate some of the language changes with Figure 2, but my two fundamental concerns remain. First, small changes in gene programs are called significant when those subtle changes may not amount to a major push towards one lineage; myogenic and chondrogenic markers are especially uninspiring. Second, there is no positive control. Without it, how good can we say the HEX pattern or any other is? If your pattern induces minor changes, does that really mean the cells have become X, Y, or Z? Not likely even if

many markers show that. There are also no comments on whether the cells are recapitulating developmental pathways.

My concerns were met with the statement that a [REDACTED] of the exercise. I got back to my statement above: I don't care about small but robust changes. I want something that will recapitulate the appropriate developmental program. The authors have not provided any response, text changes, or new experiments to assuage my concerns.

We are grateful to the reviewer for clarifying their concern. Indeed, arbitrary definitions of cell functionality defeats the purpose of having a predictive model that defines gene expression levels accurately (see our comment to Reviewer #1, Point #3). To address the reviewer's comment, we performed new experiments to generate positive controls. These positive controls used the same cell lines used in the study, but cultured with inducers of myogenic^{1,2}, osteogenic^{3,4}, chondrogenic^{5,6} and fibrotic^{7,8} differentiation. The protocols used to differentiate cells into the aforementioned musculoskeletal functionalities are well established in the field.

Gene expression of positive control or differentiated cells were compared with nanotopographically-stimulated cells by hierarchical clustering (Figure S6E-S6H, and the figure pasted below). Through this comparison, we observed that while all nanotopographies failed to stimulate cells to a fully differentiated state, specific nanotopographies induced a gene expression profile that recapitulated an early timepoint of differentiation. For example, gene expression of pre-osteoblasts and osteoblasts on NSQ were similar to day 4 of osteogenic differentiation. The text accompanying Figure 2 (page 7, starting from line 19) has now been amended to reflect the new data:

Nanotopography induces cell type-specific gene expression changes

Gene expression was used to quantitatively determine the effect of nanotopographies on cell function (Figure S5). For comparison, we differentiated the same cells cultured on conventional tissue culture plastic using established biochemical inducers of musculoskeletal differentiation (Figure S6 and see Supplementary Methods). We discuss here the changes induced by nanotopography on lineage-specific gene expression relevant to the cell type. Pre-myoblasts showed significantly higher expression of the early lineage marker MYOD1, and of the late markers MYOG and MYH7 when cultured on SQ surfaces relative to FLAT surfaces (Figure 2B-2D). This myogenic gene expression profile was similar to pre-myoblasts stimulated with biochemical inducers of myogenic differentiation for 4 days

(Figure S6E). Both pre-osteoblasts and osteoblasts showed increased expression of early (RUNX2, SP7) and late (BGLAP, SPP1) osteogenic markers when cultured on NSQ relative to FLAT (Figure 2E-2H), in line with previous studies^{7,8,10,33}. The gene expression profile of both pre-osteoblasts and osteoblasts on NSQ resembled cells osteogenically differentiated after 4 days (Figure S6F). Chondrocytes cultured on HEX showed increased expression of COL2A1 (early marker) and ACAN (late marker) compared to those cultured on FLAT (Figure 2I-2K). Chondrogenic gene expression profile induced by SQ and HEX showed the highest similarity with cells chondrogenically differentiated for 4 days (Figure S6G). Meanwhile, fibroblasts showed increased expression of pathogenic fibrosis markers, TGFB1I1, COL3A1 and ELN34, on all nanotopographies compared with FLAT (Figure 2M-2O). However, we did not observe any similarities in fibrotic gene expression profile induced by nanotopographies and fibrotic induction. In general, a cell type-specific response to nanotopography was observed. Each musculoskeletal phenotype was notably enhanced in a specific cell type and nanotopography combination compared to FLAT: SQ stimulated the myoblast phenotype, NSQ enhanced the osteoblast phenotype, HEX stimulated the chondrocyte phenotype.

Figure S6. Comparison of gene expression between cells stimulated by nanotopography and biochemical inducers of differentiation. Pre-myoblasts, pre-osteoblasts, chondrocytes and fibroblasts were differentiated with established biochemical inducers. Thereafter, gene expression from biochemically-differentiated cells were measured. (A) Myogenic, (B) osteogenic, (C) chondrogenic and (D) fibrotic differentiation were measured across multiple timepoints (n=6, 2 independent experiments). Day 0 indicates cells without biochemical induction of differentiation or undifferentiated cells. Gene expression from biochemically-differentiated cells were used as controls to determine the

extent of lineage commitment induced by nanotopography. (E-H) Hierarchical clustering of average gene expression data across all lineage-specific genes. Dendrogram shows relationship in gene expression between biochemically-stimulated and nanotopographically-stimulated cells. Notable comparisons are highlighted in blue. FLAT/SQ/NSQ/HEX denote precursor cells (i.e. pre-myoblasts, pre-osteoblasts, chondrocytes and fibroblasts) cultured on nanotopography while FLATDiff/SQDiff/NSQDiff/HEXDiff denote lineage-committed cells (i.e. myoblasts and osteoblasts) cultured on nanotopography. Gene expression from cells on nanotopography were measured after 7 days of culture, and are presented in Figure 2 of the main text.

2. My simple statement that the authors should comment on variance between cells was taken out of context stating that **[REDACTED]**

While I agree that FLAT can show the net change from baseline, it does not describe the variance in response between donors or between lines of the same cell type.

I appreciate the changes that were made but these remain concerns, especially for a more general audience journal as Nat Comm.

We may have misinterpreted the Reviewer's original comment. The Reviewer originally commented on how pre-osteoblasts and pre-myoblasts were **[REDACTED]**

Going back to this original and the latest comment, we believe that we may have misunderstood the original statement. To answer the question regarding genetic variance, we want to clarify that we used the same cell lines for pre-osteoblasts and osteoblasts (MC3T3 cell line), and for pre-myoblasts and myoblasts (C2C12). The only difference between the precursor (i.e. pre-myoblast and pre-osteoblast) and lineage committed (i.e. myoblast and osteoblast) cells (as we have now referred to them in Figure 2), is the presence of inducers of lineage commitment. Specifically, pre-osteoblasts are MC3T3 cells while osteoblasts are MC3T3 cells grown with dexamethasone, beta-glycerophosphate and ascorbic acid^{3,4}. Similarly, pre-myoblasts are C2C12 cells and myoblasts are C2C12 cells grown with horse serum^{1,2}. Indeed, to ensure a controlled system to demonstrate our morphome-based models, it was essential to keep biological heterogeneity as low as possible. The Figure 2 caption (page 10) has been appended with the following statement to clarify this:

Precursor (pre-myoblast and pre-osteoblast) and lineage committed (myoblast and osteoblast) cells were from the same cell line, but with lineage committed cells cultured in the presence of inducers of osteogenic or myogenic differentiation (see Methods for details).

Within the 7-day timepoint (at which we measure gene expression), changes at the genetic level (e.g. spontaneous mutations) are expected to be low. Previous studies have shown that gene mutation rate can go to a maximum of 2×10^{-6} per mouse lymphoblastoid cell per generation⁹. This value indicates a very low probability of mutation of each cell at each cell division event¹⁰. In our study, we observed that most cells have completely covered the entire nanotopography surface within 4 days. And within this time period, genetic variance between precursor and lineage committed cells caused by genetic instability and mutation is very probably low.

In summary, genetic variance (at the DNA level) is minimised between precursor and lineage committed cells used in this study. We hope the updated response addresses the Reviewer's concerns, and we apologise for our misunderstanding and inadequate response to the Reviewer's original comment.

References

1. Quach, N. L., Biressi, S., Reichardt, L. F., Keller, C. & Rando, T. A. Focal Adhesion Kinase Signaling Regulates the Expression of Caveolin 3 and β 1 Integrin, Genes Essential for Normal Myoblast Fusion. *Molecular Biology of the Cell* **20**, 3422–3435 (2009).
2. Clemente, C. F. M. Z., Corat, M. A. F., Saad, S. T. O. & Franchini, K. G. Differentiation of C 2C 12 myoblasts is critically regulated by FAK signaling. *American Journal of Physiology-Regulatory, Integrative and Comparative Physiology* **289**, R862–R870 (2005).
3. Quarles, L. D., Yohay, D. A., Lever, L. W., Caton, R. & Wenstrup, R. J. Distinct proliferative and differentiated stages of murine MC3T3-E1 cells in culture: an in vitro model of osteoblast development. *Journal of Bone and Mineral Research* **7**, 683–692 (1992).
4. Yan, X.-Z. *et al.* Effects of continuous passaging on mineralization of MC3T3-E1 cells with improved osteogenic culture protocol. *Tissue Engineering Part C: Methods* **20**, 198–204 (2014).
5. Perrier-Groult, E., Padeloup, M., Malbouyres, M., Galéra, P. & Mallein-Gerin, F. Control of collagen production in mouse chondrocytes by using a combination of bone morphogenetic protein-2 and small interfering RNA targeting Col1a1 for hydrogel-based tissue-engineered cartilage. *Tissue Engineering Part C: Methods* **19**, 652–664 (2013).
6. Chen, Y.-L. *et al.* Sorafenib ameliorates bleomycin-induced pulmonary fibrosis: potential roles in the inhibition of epithelial-mesenchymal transition and fibroblast activation. *Cell Death Dis* **4**, e665–e665 (2013).
7. Ji, Y.-D. *et al.* BML-111 suppresses TGF- β 1-induced lung fibroblast activation in vitro and decreases experimental pulmonary fibrosis in vivo. *Int. J. Mol. Med.* **42**, 3083–3092 (2018).
8. Negmadjanov, U. *et al.* TGF- β 1-mediated differentiation of fibroblasts is associated with increased mitochondrial content and cellular respiration. *PLoS ONE* **10**, e0123046 (2015).

9. Boesen, J. J. B., Niericker, M. J., Dieteren, N. & Simons, J. W. I. M. How variable is an spontaneous mutation rate in cultured mammalian cells? *Mutation Research/Fundamental and Molecular Mechanisms of Mutagenesis* **307**, 121–129 (1994).
10. Luria, S. E. & Delbrück, M. Mutations of bacteria from virus sensitivity to virus resistance. *Genetics* **28**, 491–511 (1943).

REVIEWERS' COMMENTS:

Reviewer #1 (Remarks to the Author):

My concerns have been addressed

Reviewer #3 (Remarks to the Author):

I would like to thank the authors and am now left with only clarifying questions.

1. In the legend of Fig S6, the authors say "Day 0 indicates cells without biochemical induction of differentiation or undifferentiated cells." I would think that these were MSCs/MC3T3/etc alone (not differentiated), so why not call them just "undifferentiated cells." The "without" is confusing; does it modify both dependent clauses at the end of the sentence?

2. Fig S6E-H are confusing relative to their legend. Are cells cultured on SQ/NSQ/HEX/FLAT with the chemical factors? Why are days mentioned in the dendrograms but not the legend? Moreover why mention FLATDiff/SQDiff/NSQDiff/HEXDiff if those are only in Figure 2???? This is very confusing. I believe that they did the positive control experiment, but its not clear if chemical induction was just done on glass (as is the standard control) or if induction occurred on patterned surfaces.

Our response to reviewers comments in blue.

REVIEWERS' COMMENTS:

Reviewer #3 (Remarks to the Author):

I would like to thank the authors and am now left with only clarifying questions.

1. In the legend of Fig S6, the authors say "Day 0 indicates cells without biochemical induction of differentiation or undifferentiated cells." I would think that these were MSCs/MC3T3/etc alone (not differentiated), so why not call them just "undifferentiated cells." The "without" is confusing; does it modify both dependent clauses at the end of the sentence?

The legend for Supplementary Figure 6 has now been amended to simply say *Day 0 indicates cells without biochemical induction of differentiation.* (We have also attached Supplementary Figure 6 below)

2. Fig S6E-H are confusing relative to their legend. Are cells cultured on SQ/NSQ/HEX/FLAT with the chemical factors? Why are days mentioned in the dendrograms but not the legend? Moreover why mention FLATDiff/SQDiff/NSQDiff/HEXDiff if those are only in Figure 2???? This is very confusing. I believe that they did the positive control experiment, but its not clear if chemical induction was just done on glass (as is the standard control) or if induction occurred on patterned surfaces.

For Supplementary Figure 6A-D, we differentiated pre-osteoblasts, pre-myoblasts, chondrocytes and fibroblast cell lines on tissue culture plastic using standard biochemical factors. At specific timepoints after induction of differentiation, we measured gene expression to obtain idea about the progression of differentiation over time. Hence, the timepoints are also present in the dendrograms.

For Supplementary Figure 6E-H we aimed to compare the gene expression between the positive control experiment and those induced by nanotopography, which is originally presented in Figure 3 (formerly Figure 2). For clarity, we have changed the labels in Supplementary Figure 6E-H to show "Control Day" to denote gene expression data from the positive control experiment from a particular timepoint. For consistency, we have changed the labels in the Supplementary Figure 6E-H to match those in Figure 2.

We apologise for the confusion regarding the figure and the accompanying legend, which has now been amended (see below).

Supplementary Figure 6. Comparison of gene expression between cells stimulated by nanotopography and biochemical inducers of differentiation. (A-D) Control experiments were performed by differentiating pre-myoblasts, pre-osteoblasts, chondrocytes and fibroblasts grown on standard tissue culture polystyrene with established biochemical inducers. Thereafter, gene expression from the control experiments were measured at different timepoints. Day 0 indicates cells undifferentiated cells. (A) Myogenic, (B) osteogenic, (C) chondrogenic and (D) fibrotic differentiation were measured across multiple timepoints (n=6, 2 independent experiments). Data are presented as mean \pm standard deviation, with individual data points presented as open faced circles. (E-H) Hierarchical clustering of average gene expression data. Dendrogram shows relationship in gene expression between biochemically-differentiated controls and nanotopographically-stimulated cells. Notable similarities in biochemically-driven differentiation (control) and nanotopography-induced gene expression are highlighted in blue. Gene expression from cells on nanotopography were measured after 7 days of culture, and are originally presented in Figure 3 of the main text.